# Theoretical Analysis and Experimental Verification of the Influence of Polarization on Counter-Propagating Optical Tweezers

**DOI:** 10.3390/mi14040760

**Published:** 2023-03-29

**Authors:** Ming Chen, Wenqiang Li, Jianyu Yang, Mengzhu Hu, Shidong Xu, Xunmin Zhu, Nan Li, Huizhu Hu

**Affiliations:** 1State Key Laboratory of Modern Optical Instrumentation, College of Optical Science and Engineering, Zhejiang University, Hangzhou 310027, China; 2Quantum Sensing Center, Zhejiang Lab, Hangzhou 311121, China

**Keywords:** optical tweezers, T-matrix, polarization

## Abstract

Counter-propagating optical tweezers are experimental platforms for the frontier exploration of science and precision measurement. The polarization of the trapping beams significantly affects the trapping status. Using the T-matrix method, we numerically analyzed the optical force distribution and the resonant frequency of counter-propagating optical tweezers in different polarizations. We also verified the theoretical result by comparing it with the experimentally observed resonant frequency. Our analysis shows that polarization has little influence on the radial axis motion, while the axial axis force distribution and the resonant frequency are sensitive to polarization change. Our work can be used in designing harmonic oscillators which can change their stiffness conveniently, and monitoring polarization in counter-propagating optical tweezers.

## 1. Introduction

Optical tweezers, which take advantage of the interaction between light and matter, are a method of observing and controlling levitated mesoscopic objects. Since being proposed by Ashkin in 1970 [1], they have been important tools in biological [2], material [3], and quantum physical [4] studies. The absence of mechanical support grants them good thermal isolation and large degrees of freedom. Optical tweezers in vacuums offer even better decoupling from the environment. They have great potential in fundamental science and high-precision sensing, for example, in verifying the inverse square law of gravity [5] and ultimate weak-force sensing [6].

The trap scheme of optical tweezers includes mainly single-beam and dual-beam counter-propagating setups [7]. Single-beam optical tweezers are suitable for levitating nanometer-sized objects. They usually need a large numerical aperture (NA) lens to produce enough gradient force to cancel the scattering force, typically NA≥0.8. As a result, they usually have a short trapping distance between the focus point and the surface of the lens or objective, typically a few millimeters. Counter-propagating optical tweezers, though, cause some difficulty in alignment, have less restriction on NA. Therefore, they can levitate a heavier object at a longer distance, which benefits measuring and controlling. In detail, a levitated object with greater mass has higher acceleration sensitivity, and a longer trapping distance indicates more spatial range for the movement of the sphere and the space for other devices such as electrodes.

To investigate the dynamics of the levitated objects, it is important to calculate the interaction force between light and matter in different conditions. The numerical method of optical tweezers includes the geometric optics method [8], the dipole approximation method [9], the FDTD method [10], and the T-matrix method [11]. The geometric method is used for levitated objects with sizes larger than the optical wavelength. It decomposes the trap beam into infinitely thin beams that behave like plane waves traveling in straight lines in a homogeneous medium. The dipole approximation method, on the contrary, is used for objects with sizes smaller than the optical wavelength. It approximates the microsphere as a dipole and calculates the force through electromagnetic field theory. The finite-difference time-domain method (FDTD) and the T-matrix method can be applied to objects with arbitrary sizes. The T-matrix method is based on the generalized Lorenz–Mie theory. While the FDTD method, considering more physical details, is good at calculating the optical force on particles of arbitrary shapes and compositions illuminated with various beams, the T-matrix method used by this article can achieve similar results with fewer computational resources, with a regular particle shape.

In comparison to single-beam optical tweezers, a counter-propagating setup includes interference issues. Researchers have utilized cross polarization [12] and parallel polarization [13], while the state of optical trapping with arbitrary polarization has not been achieved. In this article, we numerically analyze the influence of polarization on counter-propagating optical tweezers with the T-matrix method and experimentally verify the theoretical results. We used the Jones vector [14] to define the polarization condition of the counter-propagating beams, and calculated the optical force distribution, trap stiffness, and resonant frequency with different polarization. We also changed the polarization condition by rotating a half-wavelength plate on the experiment and comparing the observed resonant frequency with the theoretical value. The result of this study can be used to design a harmonic oscillator that can conveniently change its stiffness.

## 2. Materials and Methods

### 2.1. T-Matrix Method

Based on the size of the levitated object, there are three regimes: the geometrical optics regime, the Rayleigh scattering regime, and the Mie scattering regime, with the size of the levitated object larger, smaller, and close to the optical wavelength, respectively. While the geometrical optics method [8] and the dipole approximation method [9] work well in the geometrical optics regime and the Rayleigh scattering regime, respectively, the FDTD method [10] and the T-matrix method [11] can both undertake all three regimes. The FDTD method considers more physical parameters, thus adapting to more complex conditions. The T-matrix method, developed from the general Lorenz–Mie theory, uses fewer computational resources.

The interaction between light and matter comes from the change in photon momentum and can be calculated by analyzing the field scattered by the levitated object. We can represent the incident field $U_{inc}$ by a set of base vector $\psi_{n}^{\left(inc\right)}$, in which each vector is a solution of the Helmholtz function with the mode number n.
$$U_{inc}=\overset{\infty }{\underset{n}{\sum }}a_{n}\psi_{n}^{\left(inc\right)}$$
where $a_{n}$ is the coefficient of each vector. In numerical analysis, this summation is truncated on a finite number $n_{max}$. The scattered field can also be represented by a set of base vectors $\psi_{k}^{scat}$.
$$U_{scat}=\overset{\infty }{\underset{k}{\sum }}p_{k}\psi_{k}^{\left(scat\right)}$$
with the coefficient $p_{k}$. Assuming the scattering process is linear, the relation between the scattered field and the incident field is
$$p_{k}=\overset{\infty }{\underset{n}{\sum }}T_{kn}a_{n}$$
or
P=TA

The T-matrix T only depends on the levitated object and the wavelength of the trapping laser; as a result, we can calculate it once in numerical analysis.

In a coordinate system originated on the levitated object, we can expand the ingoing field and outgoing field in ingoing and outgoing vector spherical wavefunctions (VSWFs), respectively.
$$E_{in}=\overset{\infty }{\underset{n=1}{\sum }}\overset{n}{\underset{m=-n}{\sum }}a_{nm}M_{nm}^{\left(2\right)}\left(kr\right)+b_{nm}N_{nm}^{\left(2\right)}\left(kr\right)$$
$$E_{out}=\overset{\infty }{\underset{n=1}{\sum }}\overset{n}{\underset{m=-n}{\sum }}p_{nm}M_{nm}^{\left(1\right)}\left(kr\right)+q_{nm}N_{nm}^{\left(1\right)}\left(kr\right)$$
where the VSWFs are
$$M_{nm}^{\left(1,2\right)}\left(kr\right)=N_{n}h_{n}^{\left(1,2\right)}\left(kr\right)C_{nm}\left(\theta ,\phi \right)$$
$$N_{nm}^{\left(1,2\right)}\left(kr\right)=\frac{h_{n}^{\left(1,2\right)}\left(kr\right)}{krN_{n}}P_{nm}\left(\theta ,\phi \right)+N_{n}\left(h_{n-1}^{\left(1,2\right)}\left(kr\right)-\frac{nh_{n}^{\left(1,2\right)}\left(kr\right)}{kr}\right)B_{nm}\left(\theta ,\phi \right)$$
where $h_{n}^{\left(1,2\right)}\left(kr\right)$ is the first and second kind of spherical Hankel functions, $N_{n}={\left[n\left(n+1\right)\right]}^{-1/2}$ are the normalization factors, $B_{nm}\left(\theta ,\phi \right)=r\nabla Y_{n}^{m}\left(\theta ,\phi \right)$, $C_{nm}\left(\theta ,\phi \right)=\nabla \times \left(rY_{n}^{m}\left(\theta ,\phi \right)\right)$, and $P_{nm}\left(\theta ,\phi \right)=\hat{r}Y_{n}^{m}\left(\theta ,\phi \right)$ are the vector spherical harmonics, and $Y_{n}^{m}\left(\theta ,\phi \right)$ are the normalized scalar spherical harmonics. θ is the angle with the z axis, and ϕ is the angle with the x axis of the projection on the xy plane.

$M_{nm}^{\left(1\right)}$ and $N_{nm}^{\left(1\right)}$ represent the outgoing field, and $M_{nm}^{\left(2\right)}$ and $N_{nm}^{\left(2\right)}$ represent the ingoing field. These fields are undefined at the origin. We can define the spherical vector without singularity:$$RgM_{nm}\left(kr\right)=\frac{1}{2}\left[M_{nm}^{\left(1\right)}\left(kr\right)+M_{nm}^{2}\left(kr\right)\right]$$
$$RgN_{nm}\left(kr\right)=\frac{1}{2}\left[N_{nm}^{\left(1\right)}\left(kr\right)+N_{nm}^{\left(2\right)}\left(kr\right)\right]$$

It is convenient to represent the incident field with ingoing and outgoing vectors, and represent the scattered field with outgoing fields. However, this makes the incident field and scattered field both carry momentum away from the levitated object. Thus, the pure ingoing and outgoing fields in their corresponding representations are more suitable for force calculation.

The axial optical force can then be calculated as:$$\begin{array}{l} F_{z}=\frac{2n}{c}\overset{\infty }{\underset{n=1}{\sum }}\overset{n}{\underset{m=-n}{\sum }}\frac{m}{n\left(n+1\right)}Re\left(a_{nm}^{*}b_{nm}-p_{nm}^{*}q_{nm}\right)\\ \;-\frac{1}{n+1}{\left[\frac{n\left(n+2\right)\left(n-m+1\right)\left(n+m+1\right)}{\left(2n+1\right)\left(2n+3\right)}\right]}^{\frac{1}{2}}\\ \;\times Re\left(a_{nm}a_{n+1,m}^{*}+b_{nm}b_{n+1,m}^{*}-p_{nm}p_{n+1,m}^{*}-q_{nm}q_{n+1,m}^{*}\right) \end{array}$$
and the radial optical force can be calculated by the same equation with the field rotated by π/2.

### 2.2. Experimental Setup

As Figure 1 presented, we used the half-wave plate HWP1 and the polarized beam splitter PBS1 to separate a 1064 nm laser beam into two beams. We named the transmitted beam the forward trapping laser, and the reflected beam the backward trapping laser, represented as dashed and solid lines, respectively. The forward trapping laser was a linear polarized beam in the horizontal direction, and the backward trapping laser was a linear polarized beam in the vertical direction. The two trapping beams, which were focused by the lenses L1 and L2, levitated a 10-micron-diameter SiO_2_ sphere S in a vacuum chamber. The pressure in the vacuum chamber was set to 10 mBar. The NA of L1 and L2 were 0.0673. The power of both trapping beams arriving at the sphere was 100 mW. After being scattered by the sphere, the forward trapping laser then transmitted through the polarized beam splitter PBS2 and was detected by a homemade QPD. The half-wave plate HWP2, mounted on a rotation stage, was used to control the polarization state of the trapping laser. There was also a polarimeter we used to measure the polarization of the two trapping beams which is not shown in the figure.

Ideally, the two trapping laser beams were both linear polarized beams at the sphere’s position. According to the analysis of the scattered field [15], ideally, each axis’s detection signal contained only the motion information in its direction. The calculated signal $I_{z}=I_{1}+I_{2}+I_{3}+I_{4}$ was proportional to the z axis displacement, or axial displacement, and $I_{y}=I_{1}+I_{2}-I_{3}-I_{4}$ and $I_{x}=I_{1}+I_{3}-I_{2}-I_{4}$ were y and x axis displacement, respectively, or radial displacement. When the alignment of the beam and the detector was not perfect, the power of the beam was not equally divided by the four quadrants of the QPD, and the detection signal was affected by the motion of other axes. For example, if the x axis signal $I_{x}=I_{1}+I_{3}-I_{2}-I_{4}$ has a non-zero average value, then the movement of the sphere in the z axis, which causes the change in the total power $I_{z}=I_{1}+I_{2}+I_{3}+I_{4}$, generates an additional signal superimposed on the original x axis signal based on the unbalance of the power radiated on two halves of the detection plane. Usually, the resonant frequency of the axial motion and the radial motion is different, so we can observe two resonance peaks on the PSD spectrum of one signal. To detect the axial and radial motion simultaneously, the detection alignment was adjusted for large coupling in this work.

## 3. Results

### 3.1. Numerical Analysis

The interaction force between light and matter is important information for designing the optical tweezers system. For counter-propagating optical tweezers, the optical force is affected by the polarization condition of two trapping beams. Researchers have utilized cross polarization [12] and parallel polarization [13] setups. However, the optical force generated by arbitrary polarization has not been numerically analyzed.

In order to describe the polarization condition of the counter-propagating beams, we introduce the Jones vector [14]:$$J_{n}=\left(\begin{array}{c} E_{nx}e^{i\phi_{nx}}\\ E_{ny}e^{i\phi_{ny}} \end{array}\right)$$
where n=1,2 indicates the forward or backward beam, respectively, $E_{nx}$ and $E_{ny}$ are the amplitude terms, and $e^{i\phi_{nx}}$ and $e^{i\phi_{ny}}$ are the phase terms. The counter-propagating setup utilizes the two opposite lights to cancel the unwanted scattering force and strengthen the gradient force, which requires the power of the two beams to be the same; $E_{1x}^{2}+E_{1y}^{2}=E_{2x}^{2}+E_{2y}^{2}$. We introduced the power split ratio $P_{n}$ into the normalized Jones vector:$$J_{1}=\left(\begin{array}{c} \sqrt{P_{1}}e^{i\phi_{1x}}\\ \sqrt{1-P_{1}}e^{i\phi_{1y}} \end{array}\right)$$
$$J_{2}=\left(\begin{array}{c} \sqrt{P_{2}}e^{i\phi_{2x}}\\ \sqrt{1-P_{2}}e^{i\phi_{2y}} \end{array}\right)$$

If the polarization of the two trapping beams is orthogonal, for example, setting the forward trapping beam into the linear polarized laser in the x direction and the backward trapping beam into the linear polarized laser in the y direction, then it can be described by $P_{1}=0$, $P_{2}=1$, $\phi_{1x}=\phi_{1y}=\phi_{2x}=\phi_{2y}=0$.

First, we analyzed the influence of the power split ratio. Considering the spatial symmetry, we only need to analyze the change in the power split ratio of one trapping beam. We assume the backward trapping laser to be ideal and change the power split ratio of the forward trapping laser in comparison with the orthogonal condition:$$J_{1}=\left(\begin{array}{c} \sqrt{P_{1}}\\ \sqrt{1-P_{1}} \end{array}\right)$$
$$J_{2}=\left(\begin{array}{c} 1\\ 0 \end{array}\right)$$

The calculated optical force distribution is shown in Figure 2. The simulation parameters are consistent with the experiment setup, which is a 10 μm-diameter SiO_2_ sphere levitated by two 1064 nm trapping beams with an NA of 0.0673. For optical force in the x direction and y direction to be similar in the regime of this research, we chose the x direction force to represent the radial force. Figure 2a is the radial force distribution on the radial displacement of the sphere. When the polarization of the two trapping beams is orthogonal, $P_{1}=0$, the radial force is proportional to the radial displacement in the range of ∼ 1 μm, which is called the linear region. The optical tweezers can be described by the harmonic oscillator in this region. As shown in the figure, the power split ratio only slightly affects the maximum radial force. Figure 2b is the axial force distribution on the axial displacement of the sphere. When the polarization of the two beams is orthogonal, the axial force distribution has a similar trend compared with the radial force distribution, though the linear region of the axial force is larger because of a smaller light field gradient, about ∼ 10 μm, as shown in the inserted Figure 2b. We can learn from the figure that the power split ratio has a significant influence on the axial force. A non-zero $P_{1}$ can cause interference and create multiple axial balance points, making the linear region reduce to ∼0.1 μm, a quantity related to the laser wavelength. With the $P_{1}$ increase, the force gradient, or trap stiffness, becomes larger.

Although the trapped particle size is larger than the interference period, the random motion of the center of mass is confined to a small range. In this work, the sphere was in a low-vacuum environment with pressure P=10 mBar at room temperature, T=293 K. With a typical trapping stiffness, $k\approx 1\times 10^{-4}\;\left(N/m\right)$, the root-mean-square (r.m.s.) amplitude of a trapped microsphere at thermal equilibrium is $\sqrt{r^{2}}=\sqrt{k_{B}T/k\;}\approx 6\;\mathrm{nm}$ [16], where $k_{B}$ is the Boltzmann constant. It is two orders less than ∼0.1 μm, the linear region of the z-axis optical force affected by interference. Thus, the influence of the interference on the motion of the center of mass investigated in this work is reasonable.

Then, we consider the polarization phase. There are four phase terms, while the absolute phase only translates the whole system in the z direction, which is not in the scope of this work. Taking spatial symmetry into consideration, there are only two independent variables. We fix one phase term, $\phi_{1x}=0$, and define two kinds of phase differences: the first kind is the difference between the polarization phase of two perpendicular radial directions, $\Delta \phi_{1}=\phi_{1x}-\phi_{1y}$, and the second kind is the difference between the common phase of two counter-propagating waves $\Delta \phi_{tot}=\phi_{1x}-\phi_{2x}=\phi_{1y}-\phi_{2y}$. We set the power split ratio of two beams to $P_{1}=P_{2}=\frac{1}{2}$, for the phase difference, which only affects the light which is elliptical or circular polarized:$$J_{1}=\frac{\sqrt{2}}{2}\left(\begin{array}{c} 1\\ e^{i\left(-\Delta \phi_{1}\right)} \end{array}\right)$$
$$J_{2}=\frac{\sqrt{2}}{2}\left(\begin{array}{c} e^{i\left(-\Delta \phi_{tot}\right)}\;\\ e^{i\left(-\Delta \phi_{tot}\right)} \end{array}\right)$$

Figure 3 shows the optical force distribution with different phase differences. As we can see from Figure 3a,c, the two kinds of phase difference do not affect the radial force difference much. Figure 3b shows the axial force distribution of different phase differences in the first kind. Not only did the axial balanced point translate in the axial direction, but the maximum value and gradient of the axial force also changed with the second kind of phase difference. As presented in Figure 3d, the axial force distribution translates by a distance related to the wavelength in the axial direction when the second kind of phase difference changes, with its maximum value and gradient at a balanced point unchanged.

After analyzing the force distribution with different polarization conditions, we can further calculate the trap stiffness and resonant frequency, which are more convenient for experimental observations. In the linear region, the optical tweezers can be described with the harmonic oscillator model. The trap stiffness is defined as the force gradient $k=\frac{\Delta F_{r}}{\Delta r}$, where r is the displacement of the sphere and $F_{r}$ is the optical force in the r direction. The resonant frequency can also be calculated by $f=\frac{\sqrt{k/M}}{2\pi }$.

Figure 4 is the resonant frequency or trap stiffness in the axial and radial directions in different polarization conditions. According to the force distribution calculation in the previous context, the second kind of phase difference does not change the force gradient at the balance point, so only the power split ratio and the first kind of phase difference need to be considered. We can see from the figure that radial resonant frequency is not sensitive to polarization change, as discussed above. Figure 4a shows us that as the power split ratio increases, the axial resonant frequency also increases. As we can see from Figure 4b, at the same power split ratio, the axial resonant frequency is negatively correlated with the first kind of phase difference. The radial and axial resonant frequency of orthogonal polarization is 2222 Hz and 105 Hz, respectively, the radial resonant frequency is about one order greater than the axial one. However, this radial-to-axial ratio is very sensitive to the polarization condition. The axial resonant frequency becomes larger than the radial one as the power split ratio $P_{1}$ increases to 0.014.

### 3.2. Experimental Verification

Figure 5a is a series of the PSD spectrum of the x-direction signal $I_{x}$, calculated from the QPD. The rotation angle θ corresponds to the angle of polarization of the forward trapping beam compared to the direction perpendicular to the backward trapping beam. The polarization angle θ is controlled by the rotation angle of the half-wave plate HWP2 θ=2η. Ideally, the PBS1 polarizes the forward (backward) propagating beam in the horizontal (vertical) direction. The optical axis of the HWP2 is initially consistent with the polarization direction of the forward light passing through it. When the HWP2 is rotated by η, the polarization of the two beams becomes $J_{1}=\left(\begin{array}{c} 1-\sin^{2}2\eta \\ \sin^{2}2\eta \end{array}\right)=\left(\begin{array}{c} 1-\sin^{2}\theta \\ \sin^{2}\theta \end{array}\right)$, $J_{2}=\left(\begin{array}{c} 0\\ 1 \end{array}\right)$. There are two peaks on each spectrum, which are signals originating from the radial motion and the coupling signal from the axial motion. We can see the peak with a lower resonant frequency, which is related to the radial motion and is not sensitive to the change in polarization, while the coupling signal from the axial motion at a higher frequency changes significantly. The experimental resonant frequency is extracted from Figure 5a by fitting the PSD curves near the resonant peaks with the Lorenz function [17], plotted in Figure 5b. The blue plus and magenta stars are the experimental resonant frequency of the axial and radial motion, respectively, while the lines are the calculated frequency with the polarization of the two trapping beams measured by the polarimeter, which is $J_{1}=\left(\begin{array}{c} 0.9998\\ 0.0177e^{2.7885i} \end{array}\right)$, $J_{2}=\left(\begin{array}{c} 0.2039e^{1.0755i}\\ 0.9790 \end{array}\right)$. The experimental and numerical results match well. Despite the experimental setup that attempts to achieve the orthogonal polarization condition, the non-ideality of the optical elements and misalignment prevent the fulfillment of the theoretical condition. For example, the PBSs we used in the experiment have different extinction ratios of polarization of transmission and reflection, which is about 1000:1 for transmission and 20:1 to 100:1 for reflection, according to the factory data.

## 4. Discussion

In conclusion, we calculated the optical force distribution and resonant frequency of counter-propagating optical tweezers in different polarizations. The analysis has great potential in applications. A harmonic oscillator that can change its stiffness conveniently can be designed based on this work. For example, the traditional parametric feedback cooling is mainly based on the single-beam optical trap, such that modulate the trap stiffness by changing the power of the trapping laser is convenient [18]. However, in the dual-beam counter-propagating optical tweezers, the power of the two beams is required to be equal to cancel the unwanted scattering force. This makes it difficult to adjust the optical power. We can now achieve the same feedback effect by changing the polarization condition, while keeping the power of the trapping beams constant. Moreover, the traditional parametric feedback uses the difference in axial and radial resonant frequencies to distinguish the two feedback signals in the frequency domain. However, the axial (radial) feedback signal still generates a modulation in radial (axial) motion because of its feedback mechanism, which is ignorable only when the motion of the particle has been cooled down. The parametric feedback based on the polarization modulation acts only on axial motion because the radial trap stiffness is far less sensitive to polarization. Additionally, according to the observed resonant frequency of the axial and radio frequency, one can also estimate the polarization of the trapping beams without an extra detector.

## Figures and Tables

**Figure 1 micromachines-14-00760-f001:**
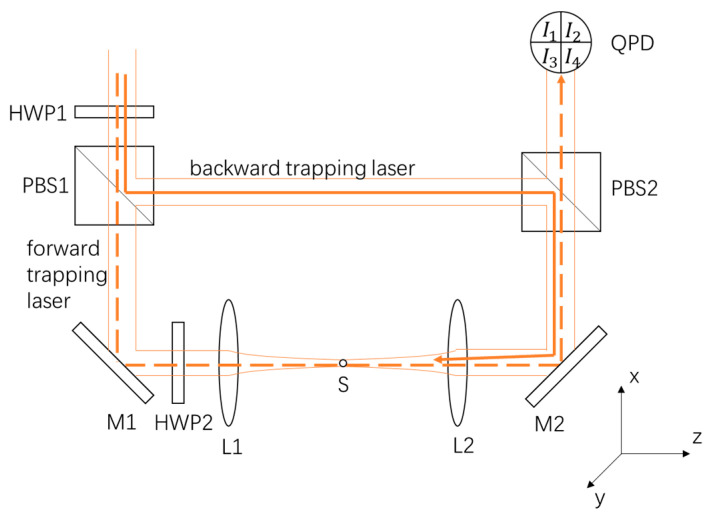
Experimental setup. The 1064 nm trapping laser was separated by a polarized beam splitter PBS1. We named the transmitted beam the forward trapping laser, and the reflected beam the backward trapping laser, indicated by the dashed line and solid line, respectively. The power of the two beams was adjusted to the same value by rotating the half-wave plate HWP1. The two beams were then aligned and focused with the numerical aperture (NA) of 0.0673. The 10-micron-diameter SiO_2_ sphere was trapped by the counterpropagating beams. The forward trapping laser, after interacting with the levitated sphere, was detected by the quadrant photoelectric detectors.

**Figure 2 micromachines-14-00760-f002:**
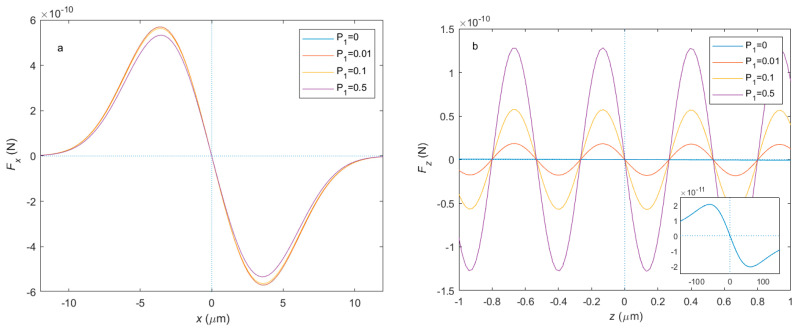
The optical force distribution of different power split ratios. (**a**) The radial force distribution and (**b**) the axial force distribution in each direction of different power split ratios. The inserted figure in (**b**) is the zoom-out axial force distribution of $P_{1}=0$.

**Figure 3 micromachines-14-00760-f003:**
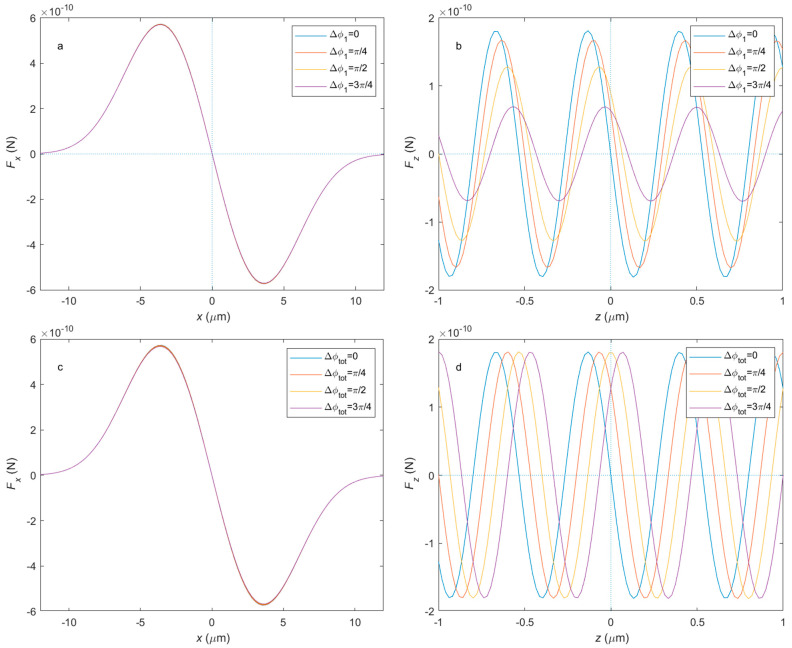
The optical force distribution of different polarization phases. (**a**) The radial force distribution and (**b**) the axial force distribution in each direction in respect to phase differences in the first kind. (**c**) The radial force distribution and (**d**) the axial force distribution in each direction in respect to phase differences in the second kind. The power split ratio is $P_{1}=P_{2}=\frac{1}{2}$.

**Figure 4 micromachines-14-00760-f004:**
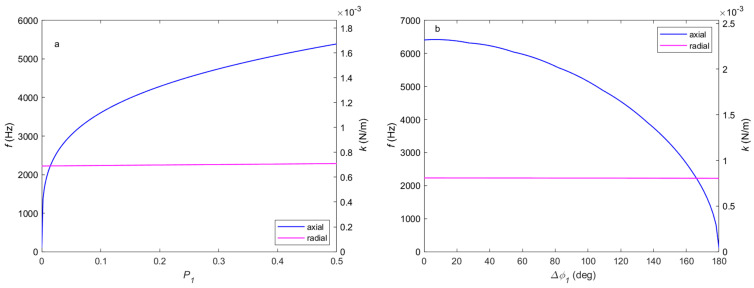
The resonant frequency or trap stiffness of different polarizations. The axial (blue) and radial (magenta) resonant frequency or trap stiffness of (**a**) different power split ratios with $P_{2}=0$ and (**b**) the first kind phase difference with $P_{1}=P_{2}=\frac{1}{2}$.

**Figure 5 micromachines-14-00760-f005:**
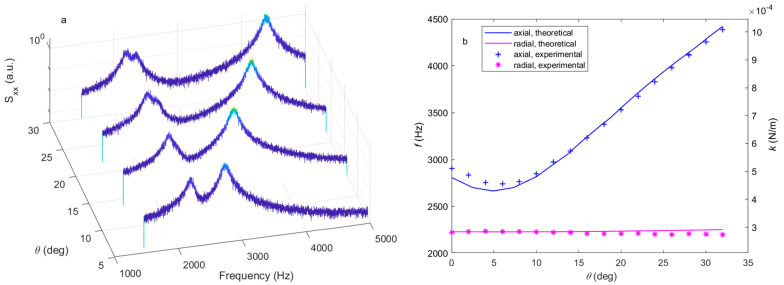
The experiment result. (**a**) The power spectrum density of the radial axis of the different polarization angles. The coupling between the radial and axial axis is enlarged to make their signal appear to be at the same spectrum. (**b**) The axial (blue) and radial (magenta) resonant frequency or trap stiffness of different angles between the polarization of two beams. The plus and star signs are experimentally detected by the QPD; the lines are theoretical calculations based on the trapping condition.

## Data Availability

Not applicable.

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
