# Peer review of "Theoretical Analysis and Experimental Verification of the Influence of Polarization on Counter-Propagating Optical Tweezers"

_micromachines, 2023, doi:10.3390/mi14040760_

Round 1

Reviewer 1 Report

The authors report numerical calculation of optical force and trapping stiffness of counter-propagating optical tweezers in arbitray polarization. The theoretical analysis is sound and verified by experimental results. While existing studies only use cross or parallel polarization in counter-propagating optical tweezers, the method of using two counter-propagating beams with different polarizations for tunable optical trapping is simple but less explored. The result has application potential such as designing harmonic oscillator with its stiffness adjustable. I agree to accept this manuscript for Micromachines.

Author Response

We would like to thank you for your careful reading, helpful comments, and constructive suggestions, which have significantly improved the presentation of our manuscript. 

Reviewer 2 Report

The authors report tunable optical trapping stiffness of counterpropagating optical tweezers by varying the polarization. The theoretical analysis is sound and verified by experimental results. The method of using two counter-propagating beams with different polarizations for tunable optical trapping is simple but less explored. I find this work very interesting. I agree to accept this manuscript for Micromachines if my following concerns are addressed.

(1) In line 33, what does the trapping distance mean? Is it really in the range of millimeters? I think the size of the trapping region of a typical optical tweezer should be in the micrometer range.

(2) The authors should check the wording of this manuscript carefully. For example, in line 95, "? is" should be "? is", "with z axis" should be "with the z axis"; in line 120, "SiO2", "2" should be subscripted; in line 189, "10μm" should be "10 μm", etc.

(3) In all figures, the y-label is not clear. It looks as if the text is cut off. In Figure 2, the Greek letter micro in the x-label is displayed strangely. The legends in Figure 3 are also not displayed correctly.

(4) Please provide more details on how the detection orientation is adjusted for the large coupling case (in line 146).

(5) What is the particle size used in the calculations? In experiments, it is said to be 10 μm in diameter. However, the period of the interference pattern in the axial direction is about half the wavelength (i.e., 0.5 μm). The trapped particle size is 20 times the interference period. My question is, when the particle is trapped in the axial direction, can the particle experience the intensity variation (interference pattern) in such a high spatial resolution, or is there an average effect of the light intensity variation? In a previous study (Optica, vol. 8 (3), pp. 409, 2021) it is said that a large particle trapped in an interference pattern mainly experiences the envelope of the light intensity rather than the interference patterns in the trapping potential well. Could the authors comment on this?

(6) It is not clear to me here from lines 187 to 189. What does a similar trend mean? Please add a figure to show the force in the axial direction in a larger area (~10 μm).

(7) It says that it is the first type in Fig. 3(a, b) and the second type in Fig. 3(c, d) (in line 197 and line 217, respectively). However, according to the definition of these two kinds of phase differences, the authors made a mistake here (reversing the two phase differences). Please check carefully. It looks like the phase difference Δ?1 should not appear in the Jones vector J2 in line 213. If the phase difference Δ?1 appears in both Jones vectors J1 and J2, then when the authors vary Δ?1, the two beams will always have the same phase and the same polarization. In this case, the authors cannot obtain the results in Figure 3(d).

(8) Line 242 says that when the polarization of two beams is orthogonal, i.e., Δ?1 = 180 deg (in Fig. 4b), the radial resonance frequency is about 5 times larger than the axial resonance frequency. However, this is not clear from Fig. 4b. Could the authors give specific values of the resonant frequency for this case?

(9) What do the different PSD curves in Fig. 5a correspond to? The caption (line 248) says "the power spectrum density of the radial axis of the different angles between the polarization of two beams". But in Fig. 5a, the label is Pn, something related to the laser power. What does Pn mean?

(10) In Fig. 5c, what does the x-axis θ mean? The caption (line 251) says that it is the angle between the polarization of two beams. However, this seems to be wrong. If the x-axis θ is the angle between the polarization of two beams, then when θ = 0 (indicating that these two beams have the same polarization), the trapping stiffness should be near the maximum, not near the minimum. I guess the θ should be the rotation angle of the half-wave plate HWP2.

(11) In line 256, it says "the different power split ratio ?1 of each spectrum is controlled by the rotation angle 256 of the half-wave plate HWP2". However, the power split ratio should be controlled by HWP1, not HWP2. HWP2 is used to rotate the linear polarization of the forward propagating beam.

(12) Line 249 says "the coupling between the radial and axial axis is increased to make their signal appear in the same spectrum". This is not clear. How do the authors increase the coupling?

(13) In line 261, "Fitting the resonant frequency of different power split ratios", does it refer to Fig. 5b? But in Fig. 5b it is θ, not the different power split ratios.

(14) In line 265, what is the value of θ when J1 and J2 are measured? According to the value of J1 and J2, it looks like the polarizations of the two beams are set to be equal.

Round 2

Reviewer 2 Report

Thank the authors for the clear response. All my concerns are solved and it can be published as it is.